# Outcomes After Complicated and Uncomplicated Mild Traumatic Brain Injury at Three- and Six-Months Post-Injury: Results from the CENTER-TBI Study

**DOI:** 10.3390/jcm9051525

**Published:** 2020-05-18

**Authors:** Daphne C. Voormolen, Marina Zeldovich, Juanita A. Haagsma, Suzanne Polinder, Sarah Friedrich, Andrew I. R. Maas, Lindsay Wilson, Ewout W. Steyerberg, Amra Covic, Nada Andelic, Anne Marie Plass, Yi-Jhen Wu, Thomas Asendorf, Nicole von Steinbüechel

**Affiliations:** 1Department of Public Health, Erasmus MC, University Medical Center Rotterdam, 3000 CA Rotterdam, The Netherlands; j.haagsma@erasmusmc.nl (J.A.H.); s.polinder@erasmusmc.nl (S.P.); e.steyerberg@erasmusmc.nl (E.W.S.); 2Institute of Medical Psychology and Medical Sociology, University Medical Center Göttingen, 37073 Göttingen, Germany; amra.covic@med.uni-goettingen.de (A.C.); annemarie.plass@med.uni-goettingen.de (A.M.P.); yi-jhen.wu@med.uni-goettingen.de (Y.-J.W.);; 3Department of Emergency Medicine, Erasmus MC, University Medical Center Rotterdam, 3000 CA Rotterdam, The Netherlands; 4Department of Medical Statistics, University Medical Center Göttingen, 37073 Göttingen, Germany; sarah.friedrich@med.uni-goettingen.de (S.F.); thomas.asendorf@med.uni-goettingen.de (T.A.); 5Department of Neurosurgery, Antwerp University Hospital, 2650 Edegem, Belgium; 6Department of Neurosurgery, University of Antwerp, 2650 Edegem, Belgium; 7Department of Psychology, University of Stirling, Stirling FK9 4LJ, UK; l.wilson@stir.ac.uk; 8Department of Biomedical Data Sciences, Leiden University Medical Center, 2333 ZC Leiden, The Netherlands; 9Department of Physical Medicine and Rehabilitation, Oslo University Hospital, 0450 Oslo, Norway; nandelic@online.no; 10Faculty of Medicine, Institute of Health and Society, Research Centre for Habilitation and Rehabilitation Models and Services (CHARM), University of Oslo, 0373 Oslo, Norway

**Keywords:** traumatic brain injury, outcome, generic and disease-specific health-related quality of life, complicated mild traumatic brain injury, functional outcome

## Abstract

The objective of this study was to provide a comprehensive examination of the relation of complicated and uncomplicated mild traumatic brain injury (mTBI) with multidimensional outcomes at three- and six-months after TBI. We analyzed data from the Collaborative European NeuroTrauma Effectiveness Research (CENTER-TBI) research project. Patients after mTBI (Glasgow Coma scale (GCS) score of 13–15) enrolled in the study were differentiated into two groups based on computed tomography (CT) findings: complicated mTBI (presence of any traumatic intracranial injury on first CT) and uncomplicated mTBI (absence of any traumatic intracranial injury on first CT). Multidimensional outcomes were assessed using seven instruments measuring generic and disease-specific health-related quality of life (HRQoL) (SF-36 and QOLIBRI), functional outcome (GOSE), and psycho-social domains including symptoms of post-traumatic stress disorder (PTSD) (PCL-5), depression (PHQ-9), and anxiety (GAD-7). Data were analyzed using a multivariate repeated measures approach (MANOVA-RM), which inspected mTBI groups at three- and six-months post injury. Patients after complicated mTBI had significantly lower GOSE scores, reported lower physical and mental component summary scores based on the SF-36 version 2, and showed significantly lower HRQoL measured by QOLIBRI compared to those after uncomplicated mTBI. There was no difference between mTBI groups when looking at psychological outcomes, however, a slight improvement in PTSD symptoms and depression was observed for the entire sample from three to six months. Patients after complicated mTBI reported lower generic and disease specific HRQoL and worse functional outcome compared to individuals after uncomplicated mTBI at three and six months. Both groups showed a tendency to improve from three to six months after TBI. The complicated mTBI group included more patients with an impaired long-term outcome than the uncomplicated group. Nevertheless, patients, clinicians, researchers, and decisions-makers in health care should take account of the short and long-term impact on outcome for patients after both uncomplicated and complicated mTBI.

## 1. Introduction

In the European Union, 1.5 million hospitalizations result from traumatic brain injury (TBI) annually [1,2]. Approximately 70–90% of patients presenting to hospital are diagnosed with mild TBI (mTBI), which is generally classified with a Glasgow Coma Scale (GCS) score of 13 to 15 [1]. Williams et al. elaborated further on this description of mTBI and proposed that intracranial abnormalities on computed tomography (CT) at presentation should be taken into account. This resulted in distinguishing patients after complicated (presence of trauma-related intracranial abnormalities and/or depressed skull fracture on CT) and uncomplicated (absence of intracranial abnormalities and/or depressed skull fracture on CT) mTBI [3]. The sensitivity of CT has improved over the years, and an abnormal scan may no longer have the same significance.

Previous research concerning the impact of complicated and uncomplicated mTBI on outcome has been contradictory. On the one hand, the presence or absence of intracranial abnormalities is seen as relevant to prognosis, since complicated mTBI has been associated with cognitive and functional outcome comparable to patients after moderate TBI (GCS 9–13) [3,4,5,6]. In contrast, other studies have shown no relationship between complicated mTBI and cognitive and functional outcome [7,8,9]. Additionally, previous research reported divergent results concerning longitudinal outcome after complicated and uncomplicated mTBI. In some studies, patients after complicated mTBI reported worse outcome than individuals after uncomplicated mTBI [7,10,11]. While others observed improvement in patients after complicated mTBI over time and showed slower recovery in the uncomplicated mTBI group [12]. A few studies even found no significant differences between the two groups [7,9].

In previous research, the focus on complicated and uncomplicated mTBI research has mainly been on neurocognitive and functional outcome, and post-concussion symptoms [4,5,7,11,13,14,15]. However, nowadays, generic, and disease-specific health-related quality of life (HRQoL) have been acknowledged as an important outcome after TBI [16,17] for both patients and clinicians. HRQoL reflects an individual’s perception of how an illness and its treatment affect the physical, mental, cognitive, and social aspects of someone’s life [16,18,19].

Despite an abundance of studies and decades of research on complicated and uncomplicated patients after mTBI, only few studies have examined HRQoL outcomes in patients after complicated and uncomplicated mTBI [20] and differences regarding outcome between these groups remain poorly understood. There has not yet been a study with a large sample size, taking longitudinal changes into account, and the ability to compare patients on different outcome domains: generic and disease specific HRQoL, functional outcome and symptomatology, such as post-traumatic stress, depression and anxiety. Moreover, the authors of the present study are not aware of any studies using a multivariate approach when exploring HRQoL, functional and psychological outcomes in adult patients [21]. As (m)TBI may affect different areas simultaneously, it is important to investigate the differences between groups by using multidimensional approaches [22].

We hypothesize that patients after complicated mTBI report lower generic and disease specific HRQoL and worse functional outcomes compared to uncomplicated mTBI at three and six months. Additionally, we do not expect considerable improvement in outcome from three to six months.

More insight is needed regarding the impact of complicated mTBI compared to uncomplicated mTBI on outcome from a longitudinal perspective. Therefore, the objective of this study was to provide a comprehensive multidimensional approach in analyzing the effects of complicated and uncomplicated mTBI on outcome in a large sample of individuals at three- and six-months post-injury.

## 2. Materials and Methods

### 2.1. Study Design

In this study, we analyzed patients who were enrolled in the Collaborative European NeuroTrauma Effectiveness Research (CENTER-TBI) research project. This is a multicenter, prospective, observational, longitudinal cohort study, which was conducted in Europe and Israel [1,23]. Patients with all severities of TBI who presented to hospital were included between 19 December 2014 and 17 December 2017. A clinical diagnosis of TBI, an indication for a CT scan, presenting to a center within 24 h after the injury, and informed consent were used as inclusion criteria. Informed consent, adhering to local and national requirements, had to be obtained prior to inclusion, either personally or through a legally acceptable representative. At any point in time during the study, patients were free to withdraw without stating a reason. Patients with severe pre-existing neurological disorders, which could invalidate assessment of outcomes, were excluded. Patients were recruited in three strata: emergency room (ER; patients evaluated at the ER and discharged afterwards), admission (ADM; patients admitted to hospital ward), and intensive care unit (ICU; patients who were primarily admitted to the ICU) [23]. The main descriptive findings of CENTER-TBI can be found in Steyerberg et al. [24].

### 2.2. Study Participants

In the current study analysis, participants were included if they had mTBI (GCS 13–15). They were differentiated into two groups: complicated and uncomplicated mTBI. Complicated mTBI was specified as the presence of any traumatic intracranial abnormalities on first CT scan and uncomplicated was specified as absence of any traumatic intracranial abnormality on first CT. The presence of intracranial abnormality was defined as the detection of at least one of the following twelve findings on CT scan: mass lesion, extra axial hematoma, epidural hematoma, acute or chronic subdural hematoma, one or multiple subdural collections/mixed density hematomas, contusion, traumatic axonal injury (TAI), traumatic subarachnoid hemorrhage, intraventricular hemorrhage, midline shift, or cisternal compression. In the current study, a linear or depressed skull fracture, in the absence of structural intracranial abnormalities, was not considered as a criterion for complicated mTBI, however, this has been used as a determinant of intracranial abnormalities in previous research. Furthermore, participants had to be ≥16 years of age and classified as GOSE ≥3.

The American Society of Anesthesiologists Physical Status Classification System (ASA-PS) was used to assess individuals’ health status before the injury [25].

For all analyses, a complete case analysis for the following variables at three and six months was performed: SF-36, QOLIBRI, GOSE, PCL-5, PHQ-9, and GAD-7. For more details, see Figure 1.

### 2.3. Instruments

Outcome was assessed as a multidimensional construct by using seven instruments measuring generic and disease-specific HRQoL, functional outcome, and psycho-social domains including post-traumatic stress, depression, and anxiety. The following measures were analyzed:

#### 2.3.1. Health-Related Quality of Life (HRQoL)

Generic HRQoL. The 36-item Short Form (SF-36v2) Health Survey is a multidimensional self-report questionnaire measuring the subjective health state including physical, mental, and social functioning [26]. The questionnaire comprises 36 items covering eight domains and applies different response scales from a dichotomous (“yes”/”no”) to a polytomous five-point Likert scale. For more information see Ware et al. [26]. For our analyses, we used the two summary component scores: the physical component summary (PCS) and mental component summary (MCS) score, which measure physical functioning and mental health, respectively. The total score of the SF-36 ranges from 0 to 100, whereby higher values indicate higher HRQoL, and total scores below 40 are considered impaired [27].

Disease specific HRQoL. The Quality of Life after Brain Injury (QOLIBRI) is a 37-item self-report instrument that measures level of satisfaction with various aspects of TBI-specific HRQoL [28]. It entails six scales evaluating key aspects of life: the first four scales assess ‘satisfaction’ with cognition, self, daily life and autonomy and social relationships and the last two scales measure ‘feeling bothered’ with emotions and physical problems. Responses are given on a five-point Likert scale which extends from 0 (“not at all”) to 4 (“very”). The total score was transformed linearly to range from 0 to 100; higher values indicate better HRQoL [28]. For the QOLIBRI, scores below 60 are considered impaired [29].

#### 2.3.2. Functional Outcome

Glasgow Outcome Scale—Extended (GOSE) measures functional outcome after TBI. Functional outcome is rated by a clinician on an eight point scale: 1 (dead), 2 (vegetative state), 3 (lower sever disability), 4 (upper severe disability), 5 (lower moderate disability), 6 (upper moderate disability), 7 (lower good recovery) and 8 (upper good recovery) and is derived from eight questions [30]. In the present study, functional impairment was classified as a GOSE score ≤6 [31]. Structured interviews and self-report questionnaires were used to collect the GOSE. A multistate model was used to impute the 180-day GOSE when patients scores were outside the 5- to 8-month window (msm R package) [32]. Patients with GOSE 1 (dead) were excluded from analyses, and GOSE levels 2 and 3 were collapsed into one.

#### 2.3.3. Post-Traumatic Stress, Depression, and Anxiety

The Post-traumatic Stress Disorder Checklist-5 (PCL-5) [33] measures 20 symptoms of post-traumatic stress disorder (PTSD) based on the Diagnostic and Statistical Manual of Mental Disorders, 5th edition (DSM-V) [34] by using a five-point scale (from 0 “not at all” to 4 “extremely”). The total score ranges from 0 to 80, whereby higher values indicate greater impairment and a score of ≥33 is considered indicative of clinically relevant PTSD [35].

The Patient Health Questionnaire (PHQ-9) [36] is a nine item self-assessment instrument evaluating depression symptoms in the past two weeks using a four-point Likert scale (from 0 “not at all” to 3 “nearly every day”) based on DSM-IV criteria [37]. The maximum score is 27 and the higher the score, the greater the indication for depressive symptoms. A score of ≥10 is seen as a strong indication for clinically relevant depressive symptoms and cutoffs of 5, 10, 15 and 20 indicate mild, moderate, and moderately severe to severe depressive symptoms, respectively [36,38].

The Generalized Anxiety Disorder questionnaire (GAD-7) [39] measures anxiety symptoms in the past two weeks by using seven items with a four-point scale (from 0 “not at all” to 3 “nearly every day”). The total score ranges from 0 to 27. A score of ≥10 is generally seen as an indicator for the presence of anxiety disorder and cutoffs 5, 10 and 15 indicate mild, moderate, and severe anxiety, respectively [39].

### 2.4. Ethical Approval

The CENTER-TBI study (EC grant 602150) was conducted in accordance with all relevant laws of the EU if directly applicable or of direct effect, and all relevant laws of the country where the recruiting sites were located, including, but not limited to, the relevant privacy and data protection laws and regulations (the “Privacy Law”), the relevant laws and regulations on the use of human materials, and all relevant guidance relating to clinical studies from time to time in force including, but not limited to, the ICH Harmonised Tripartite Guideline for Good Clinical Practice (CPMP/ICH/135/95) (“ICH GCP”) and the World Medical Association Declaration of Helsinki entitled “Ethical Principles for Medical Research Involving Human Subjects”. Ethical approval was obtained for each recruiting site. Informed Consent was obtained for all patients recruited in the Core Dataset of CENTER-TBI and documented in the e-CRF. The list of sites, Ethical Committees, approval numbers, and approval dates can be found on the official Center TBI website www.center-tbi.eu/project/ethical-approval.

### 2.5. Statistical Analyses

For all analyses, data was extracted from the INCF Neurobot tool (INCF, Solna, Sweden), a clinical study data management tool. Core data set version 2.1 (data frozen in January 2019) was used for all analyses in this manuscript. Descriptive analyses for care paths, demographic and socio-economic characteristics, pre-injury health status and medical history, cause of injury, clinical presentation, CT characteristics, SF-36 MCS and SF-36 PCS, PCL-5, PHQ-9, GAD-7, QOLIBRI, and GOSE were performed and analyzed for patients with complicated and uncomplicated mTBI at three and six months post-injury. Descriptive statistics show the number (*N*) and percentages (%) for categorical variables and median and interquartile range (*IQR*) for continuous and ordinal variables. To compare individuals after complicated mTBI to uncomplicated mTBI, Chi-square tests and Mann–Whitney *U*-tests were used for categorical and continuous variables, respectively.

We compared individuals with a least one impaired outcome with those classified as not impaired at all (i.e., each outcome value did not exceed respective cut-off value). For analyses of individuals who were classified as impaired according to the cut-off values of unfavorable outcome for each outcome variable, new variables with classification (impaired vs. not impaired) were calculated for each dependent variable separately. See the instrument descriptions for the respective selected cut-off values. Distributions of individuals classified as reporting impaired outcome within mTBI groups were compared by using Chi-square tests for both three and six months after TBI.

Data were analyzed using a multivariate repeated measures approach (MANOVA-RM), suitable for non-normal data with covariance heterogeneity, to provide robust test statistics [40,41]. The outcome construct (dependent variables) consisted of seven instruments assessing outcomes (SF-36 MCS and SF-36 PCS, PCL-5, PHQ-9, GAD-7, QOLIBRI, and GOSE). The between effect was defined by complicated and uncomplicated mTBI groups. The within effect was defined by time points (three and six months after mTBI).

For post hoc comparisons, we used repeated measures ANOVAs for non-normal data and significance was assessed at α < 0.007, applying a Bonferroni-adjustment ($\alpha_{adj}=\frac{0.05}{7}$).

All analyses were conducted using R version 3.6.1 [42] with application of the MANOVA-RM package [40] for both MANOVA-RM and post-hoc repeated measures ANOVAs. Appendix A gives a detailed overview of the methodology. The significance level was determined as α < 0.05 for Chi-square tests, Mann–Whitney *U*-tests, and multivariate analysis and α < 0.007 for post hoc comparisons between groups.

## 3. Results

### 3.1. Study Sample

The total CENTER-TBI cohort included 2955 patients after mTBI and our study sample consisted of 1104 patients (37.4%) who were interviewed at both time points (3- and 6-months after mTBI) and completed all seven outcomes. Included patients were admitted to the ER (30.6%), ADM (47.6%), or the ICU (21.8%) and had sustained either uncomplicated mTBI (48.5%) or complicated mTBI (51.5%). The mean age of individuals after mTBI was 52.3 years (SD = 18.8) and 63.4% were male. The majority were injured by a fall (47%) or traffic incident (39%) and approximately 11% reported having experienced a TBI previously. Patients after complicated mTBI were significantly older (mean age: 54.5 vs. 50.3) compared to those after uncomplicated mTBI and were less likely to be classified as “working”, however, they less often reported a previous TBI (*p* < 0.01). Patients after complicated mTBI were more often classified with a GCS score of 13 and 14. For more details, see Table 1.

### 3.2. Descriptive Statistics of Outcomes

Figure 2 provides an overview on outcome instruments and uncomplicated and complicated mTBI groups and time points. Patients after complicated mTBI reported both lower generic (PCS and MCS scores) and lower disease-specific HRQoL (QOLIBRI), lower functional outcome (GOSE), higher PCL-5, PHQ-9, and GAD-7 scores compared to individuals after uncomplicated mTBI at both time points. In general, small differences were observed for both groups between 3 and 6 months.

Appendix B provides mean values (M) and standard deviations (SD) for the seven outcomes clustered in the two mTBI groups and time points.

Patients were considered impaired when the corresponding cut-offs were reached. The percentage of impaired individuals (i.e., with at least one of the seven outcomes being impaired) in the total sample was 53% at three months and 49% at six months after TBI. In the uncomplicated mTBI group, 51% and 49% were classified as impaired at three and six months, respectively. In the complicated mTBI group, 53% at three months and 48% at six months had at least one impaired outcome. For development of impaired outcomes in individuals after uncomplicated and complicated mTBI at three and six months, see Appendix C.

For both mTBI groups, there were patients included in our study sample who could be considered as reporting impaired outcomes. Therefore, these individuals with impaired outcomes were inspected separately. Table 2 gives an overview for each dependent variable by uncomplicated and complicated mTBI groups at three- and six-months post-injury.

At three months post TBI, there were significantly more individuals with impaired outcomes after complicated mTBI according to SF-36 MCS score, QOLIBRI, and GOSE (*p* < 0.05). Concerning the GOSE scale, 41% of patients after complicated mTBI were considered to be impaired (cut-off value ≤ 6) and 23% of the individuals after uncomplicated mTBI. Additionally, more patients with impaired generic (SF-36 MCS < 40) and disease-specific (QOLIBRI < 60) HRQoL were observed within the complicated mTBI group (for details, see Table 2).

At six months after TBI, we observed significantly more individuals with impaired outcomes after complicated mTBI according to the GOSE scale (*p* < 0.05). More than one third of the patients after complicated mTBI showed impairments on the GOSE, compared to 16% of individuals after uncomplicated mTBI. Among the other outcomes, the distribution within mTBI group was equal.

### 3.3. MANOVA-RM

MANOVA-RM showed significant differences between complicated and uncomplicated mTBI groups as well as between time points. No significant interaction was found between both main effects. There were significant differences between mTBI groups in three (generic and disease-specific HRQoL and functional outcome) out of seven outcomes. All but one domain (i.e., anxiety) differed between both time points (see Table 3 for test statistics).

### 3.4. Post-Hoc Comparisons

Table 4 provides an overview on differences between mTBI groups, time points, and interaction between both effects according to the results of ANOVA-RM.

Post-hoc comparisons revealed, in contrast to the multivariate results, a more detailed insight into differences between the main effects and their interaction for each dependent variable. Therefore, differences between mTBI groups and time points for the seven outcomes are reported separately. It is important to note that, with the exception of the mean depression score in the complicated mTBI group, all other outcomes were on average above the clinically relevant cut-off points.

#### 3.4.1. Health-Related Quality of Life

Generic HRQoL. The PCS (SF-36) showed a significant within effect with an increase in HRQoL from three (M = 46.52, SD = 10.43) to six months (M = 48.10, SD = 10.21) after TBI.

The MCS (SF-36) revealed significant differences between both the mTBI groups and the observed time points. Individuals after uncomplicated mTBI (M = 48.43, SD = 11.01) reported significantly higher HRQoL compared to those after complicated mTBI (M = 46.61, SD = 12.07). There was a slight but significant increase of MCS score from three (M = 47.14, SD = 11.57) to six months (M = 47.95, SD = 11.56) after mTBI.

Disease-specific HRQoL. For the QOLIBRI, significant differences between both mTBI groups and time points were reported. HRQoL was significantly higher in the uncomplicated mTBI group (M = 74.97, SD = 16.82) compared to the complicated mTBI group (M = 71.56, SD = 17.33). In addition, an increase of HRQoL was observed between three (M = 72.84, SD = 17.05) and six months (M = 73.79, SD = 17.24) following mTBI for both groups.

#### 3.4.2. Functional Outcome

For the GOSE, significant differences in both the between and within effects were detected. Patients after complicated mTBI showed significantly higher disability levels (M = 6.63, SD = 1.37) compared to those with uncomplicated mTBI (M = 7.24, SD = 1.08). A significant increase concerning recovery was observed for both groups from three (M = 6.86, SD =1.301) to six months (M = 7.02, SD = 1.23).

#### 3.4.3. Post-Traumatic Stress, Depression, and Anxiety

Results showed a significant decrease of PTSD related symptoms (PCL-5) from three (M = 12.54, SD = 13.44) to six months (M = 11.41, SD = 12.98) after mTBI. Depression related symptoms measured by the PHQ-9 were slightly but significantly higher for three (M = 4.93, SD = 5.04) compared to six months (M = 4.61, SD = 5.01) after mTBI. Anxiety measured by GAD-7 revealed no significance either between mTBI groups or between time points.

## 4. Discussion

This study focused on the outcome of patients, included in a large European database, after complicated and uncomplicated mTBI at three- and six-months post-injury. The results may contribute significantly to existing literature concerning this topic. A better understanding of the relation between complicated and uncomplicated mTBI and different outcomes may improve intervention strategies and the recovery process of patients after mTBI.

When comparing individuals after complicated and uncomplicated mTBI, patients after complicated mTBI reported lower generic and disease specific HRQoL and worse functional outcome at both time points. Nevertheless, we did not find considerable improvement in outcomes from three to six months as mean scores on all outcome measures were within average range and not classified as impaired. The mean PHQ-9 score for patients after complicated mTBI at three months post-injury reached the cut-off for mild depressive symptoms, which indicates the need for an appropriate follow-up and early treatment. It is noteworthy that the number of individuals with impaired functional and psychological outcomes was considerably higher in the complicated mTBI group than in the uncomplicated group, especially three months post-injury. Therefore, one should ensure that these patients are detected early, and their health status should be monitored longitudinally, to provide targeted and timely treatment. Ultimately, the differences between some measures were small and significant baseline differences between the groups might contribute to the reported findings. Especially, when keeping in mind that patients after complicated mTBI were often more severely injured according to the GCS score.

In previous research, nearly 40% of patients with mTBI reported depressive symptoms within three to six months post-injury [43]. In addition, 33% of patients with mTBI were functionally impaired three months post-injury [13]. When specifically focusing on individuals after complicated and uncomplicated mTBI, contradictory results have been reported regarding the impact of complicated and uncomplicated mTBI on outcome. Ponsford et al. found that the majority of individuals after uncomplicated mTBI make a good recovery on average 7 months post-injury, when focusing on post-concussion symptoms, depression/anxiety and mental and physical quality of life scores [44]. In various studies, complicated mTBI is seen as a key component when predicting outcome [3,4,5,6]. However, other research did not find any relation between functional outcome and complicated mTBI [7,8,9]. Furthermore, HRQoL has been under-investigated in previous research and a multidimensional approach to outcome assessment such as in this study has been lacking.

The present study is novel because there has not yet been a study with such a large sample size and simultaneously having the ability to compare patients on different outcome levels such as generic and disease-specific HRQoL, functional outcome, and symptomatology. Moreover, the methodology used in this study strengthened the results, since a multivariate statistical approach suitable for non-normal distributed data with less assumptions and restrictions as in most other research was applied. In particular, the method combines information from multiple outcome measures and is also suitable for non-normal data.

Several limitations concerning this study should be taken into account. It is important to note that the effect sizes were small for the PCS and MCS scores and PHQ-9, and statistical significance was most likely due to the sample size used in this study. Resilience and coping, which was not measured in this study, might have impacted outcome. Maestas et al. reported that pre-injury coping in the sense of strengthening resilience could impact outcome after uncomplicated and complicated mTBI as coping may impact resilience [20]. Williams’ classification of complicated versus uncomplicated may underestimate the presence and type of TBI abnormalities as routine magnetic resonance imaging has proven to be far more sensitive and is the preferable tool [45]. Generalizability of the results presented in this study is restricted since adjustment for baseline covariates between the two groups was not provided. In the current study implications of treatment after mTBI have not been accounted for, which could influence the course of recovery after mTBI. Lastly, patients who are still experiencing lower HRQoL and lower functional outcome might have been more likely to participate, resulting in response bias [46].

For future research it would be interesting to look at outcome at later follow-up times such as five to ten years post-TBI. In addition, return to work or school after complicated and uncomplicated mTBI should be assessed since this could influence subjective well-being [47], and has a major impact on societal costs [1]. Furthermore, research into biomarkers and localization of the abnormality on CT or magnetic resonance imaging scans can refine the conclusions drawn in this study [1,10]. Lastly, to establish a better understanding considering outcome after complicated and uncomplicated mTBI, outcomes in this study should be compared to patients with non-brain injured trauma as well as the general population, and complicated and uncomplicated mTBI groups should be further differentiated by GCS score.

## 5. Conclusions

To conclude, the present results indicate that patients after complicated mTBI reported lower generic and disease specific HRQoL and worse functional outcome compared to patients after uncomplicated mTBI at three and six months. However, differences between some measures were small and there were significant baseline differences between the groups that might contribute to the findings. Both groups showed a tendency to improve on outcome from three to six months after TBI. Additionally, the complicated mTBI group comprised more patients with impaired outcomes than the uncomplicated group. Considering this, patients, clinicians, researchers, and decisions-makers in health care should be taking the short and long-term effects on outcome for patients after both uncomplicated and complicated mTBI into account. At the same time, individually tailored therapy should be provided early on for those who show deficiencies in recovery, HRQoL, psychological and psychosocial outcomes.

## Figures and Tables

**Figure 1 jcm-09-01525-f001:**
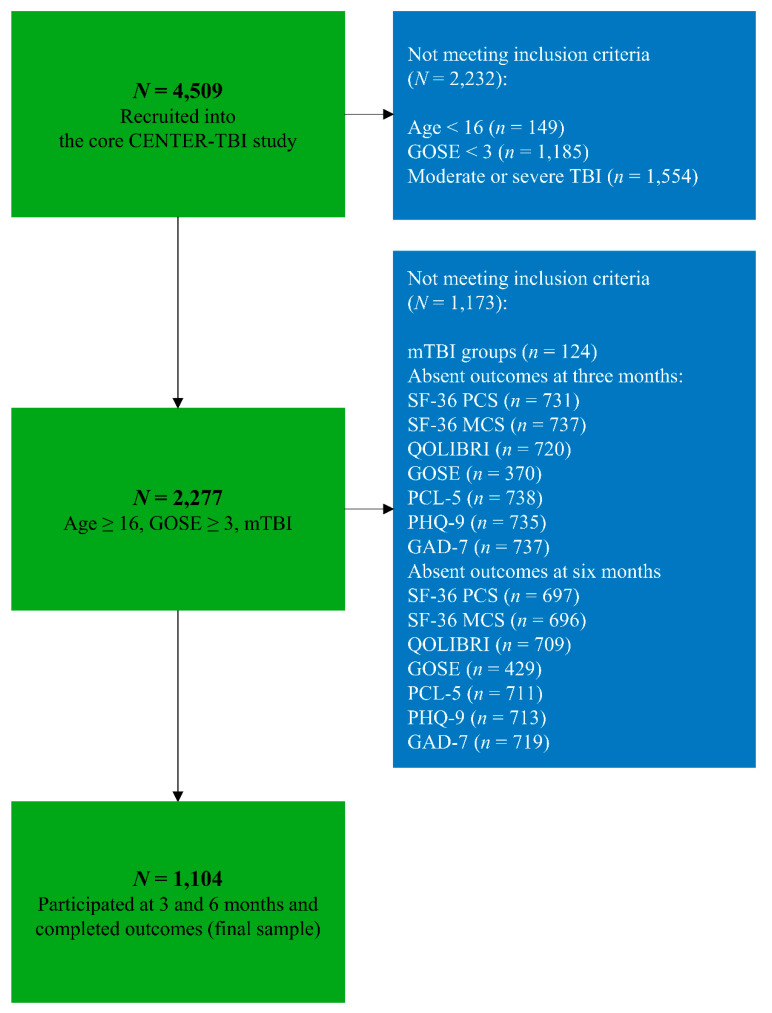
Flowchart sample size. Abbreviations: *N* = number; mTBI = mild traumatic brain injury; GOSE = Glasgow Outcome Scale—Extended; TBI = traumatic brain injury; SF-36 PCS = Short Form (36) Health Survey (physical component score); SF-36 MCS = Short Form (36) Health Survey (mental component score); QOLIBRI = Quality of Life after Brain Injury; PCL-5 = Posttraumatic Stress Disorder Checklist; PHQ-9 = Patient Health Questionnaire; GOSE = Glasgow Outcome Scale—Extended; GAD-7 = Generalized Anxiety Disorder questionnaire

**Figure 2 jcm-09-01525-f002:**
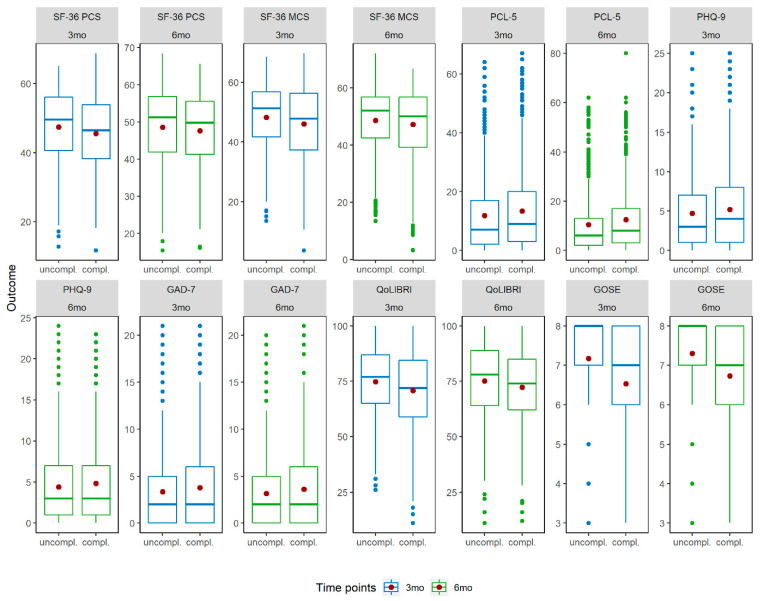
Boxplots for outcomes by time points and uncomplicated and complicated mTBI groups. Note: The Y-axis of the boxplots are adapted to the scales of the outcomes. Red dots within boxplots indicate the mean value. Abbreviations. mTBI = mild traumatic brain injury; SF-36 PCS = Short Form (36) Health Survey (physical component score); SF-MCS = Short Form (36) Health Survey (mental component score); PCL-5 = Posttraumatic Stress Disorder Checklist; PHQ-9 = Patient Health Questionnaire; GAD-7 = Generalized Anxiety Disorder questionnaire; QOLIBRI = Quality of Life after Brain Injury; GOSE = Glasgow Outcome Scale—Extended; 3mo = 3 months; 6mo = 6 months.

**Table 1 jcm-09-01525-t001:** Characteristics of the study sample.

	Study Sample	Uncomplicated	Complicated	*p*-Value
*N*	1104	569 (51.5%)	535 (48.5%)	
*Care paths*				<0.01
**ER**	338 (30.6%)	286 (50.3%)	52 (9.7%)	
**Admission**	525 (47.6%)	236 (41.5%)	289 (54.0%)	
**ICU**	241 (21.8%)	47 (8.3%)	194 (36.3%)	
*Demographic characteristics*				
**Gender** (male)	700 (63.4%)	352 (61.9%)	348 (65.0%)	0.272
**Age**^1^ (years)	54 [37.3–67]	51 [35–65]	58 [40–68]	<0.01
*Socio-economic characteristics*				
**Education**^1^ (years)	14 [12–17]	14 [12–17]	14 [11–17]	0.054
**Employment status before injury**				
Working ^2^	593 (53.7%)	331 (58.2%)	262 (49.0%)	<0.01
*Pre-injury health status and medical history*				
**Pre-injury ASA-PS classification**				0.175
A patient with mild systemic disease	377 (34.1%)	181 (31.8%)	196 (36.6%)	
A patient with severe systemic disease	106 (9.6%)	60 (10.5%)	46 (8.6%)	
**Previous TBI**	129 (11.7%)	84 (14.8%)	45 (8.4%)	<0.01
*Cause of injury*				
**Injury Mechanism**				0.409
Road traffic accident	429 (38.9%)	222 (39.0%)	207 (38.7%)	
Incidental fall	517 (46.8%)	265 (46.6%)	252 (47.2%)	
Other non-intentional	60 (5.4%)	36 (6.3%)	24 (4.5%)	
Violence/assault	38 (3.4%)	19 (3.3%)	19 (3.6%)	
Act of mass violence	1 (0.1%)	-	1 (0.2%)	
Suicide attempt	7 (0.6%)	2 (0.4%)	5 (0.9%)	
Other	36 (3.3%)	20 (3.5%)	16 (3.0%)	
Unknown	16 (1.4)	5 (0.9%)	11 (2.1%)	
*Clinical presentation*				
**GCS baseline** ^1^	15 [15–15]	15 [15–15]	15 [14,15]	<0.01
13	65 (5.9%)	9 (1.6%)	56 (10.5%)	
14	155 (14.0%)	53 (9.3%)	102 (19.1%)	
15	884 (80.1%)	507 (89.1%)	377 (70.5%)	
*CT characteristics*				
**Computed Tomography**				
Any intracranial injury on first CT	535 (48.5%)	0 (0.0%)	535 (100%)	<0.01

^1^ Data are displayed as median, with the first and third quartile given within brackets. ^2^ Working = working 35 h or more per week; working 20–34 h per week; working less than 20 h per week and/or special employment/sheltered employment. Abbreviations: mTBI: mild traumatic brain injury; ER: emergency room; ICU: intensive care unit; ASA-PS: The American Society of Anaesthesiologists (ASA) Physical Status Classification System; TBI: traumatic brain injury; GCS: Glasgow Coma Scale; CT: Computed Tomography.

**Table 2 jcm-09-01525-t002:** Percentages of impaired individuals according to the respective cut-off values.

Outcome	Time Points	Uncomplicated mTBI (*n* = 569)	Complicated mTBI (*n* = 535)
SF-36 PCS	3mo	24%	29%
	6mo	21%	22%
SF-36 MCS	3mo	22%	30%
	6mo	21%	28%
PCL-5	3mo	10%	10%
	6mo	8%	10%
PHQ-9	3mo	16%	19%
	6mo	16%	16%
GAD-7	3mo	11%	13%
	6mo	8%	11%
QOLIBRI	3mo	19%	26%
	6mo	19%	21%
GOSE	3mo	23%	41%
	6mo	16%	35%

Abbreviations: mTBI = mild traumatic brain injury; 3mo = 3 months; 6mo = 6 months; SF-PCS = Short Form (36) Health Survey (physical component score); SF-MCS = Short Form (36) Health Survey (mental component score); PCL-5 = Posttraumatic Stress Disorder Checklist; PHQ-9 = Patient Health Questionnaire; GAD-7 = Generalized Anxiety Disorder questionnaire; QOLIBRI = Quality of Life after Brain Injury; GOSE = Glasgow Outcome Scale—Extended; *n* = number of cases. Note. Cut-off values: SF-36 PCS and SF-36 MCS < 40, PCL-5 ≥ 33, PHQ-9 ≥ 10, GAD-7 ≥ 10, QOLIBRI < 60, GOSE ≤ 6.

**Table 3 jcm-09-01525-t003:** Results of repeated measures MANOVA and repeated measures ANOVA.

Analysis	Dependent Variable(s)	Independent Variable	(M)ATS	df1	df2	*p*
MANOVA RM	Multiple outcomes *	mTBI	197.538	-	-	**<0.001**
		Time points	34.708	-	-	**<0.001**
		mTBI: Time points	2.932	-	-	0.158
ANOVA RM	SF-36 PCS	mTBI	5.897	1	1365.422	0.015
		Time points	61.133	1	-	**<0.001**
		mTBI: Time points	4.361	1	-	0.037
	SF-36 MCS	mTBI	7.879	1	1399.985	**0.005**
		Time points	10.502	1	-	**0.001**
		mTBI: Time points	3.058	1	-	0.080
	PCL-5	mTBI	5.481	1	1366.071	0.019
		Time points	16.902	1	-	**<0.001**
		mTBI: Time points	0.653	1	-	0.448
	PHQ-9	mTBI	2.632	1	1386.136	0.114
		Time points	9.075	1	-	**0.005**
		mTBI: Time points	0.032	1	-	0.848
	GAD-7	mTBI	3.216	1	1425.187	0.073
		Time points	3.137	1	-	0.077
		mTBI: Time points	0.026	1	-	0.872
	QOLIBRI	mTBI	12.25	1	1337.174	**<0.001**
		Time points	8.588	1	-	**0.003**
		mTBI: Time points	2.980	1	-	0.084
	GOSE	mTBI	80.944	1	1444.067	**<0.001**
		Time points	26.15	1	-	**<0.001**
		mTBI: Time points	1.057	1	-	0.304

Note: TBI severity = between effect (uncomplicated and complicated mTBI); Time = within effect (time points 3 and 6 months after TBI); *p* = *p*-value based on parametric bootstrapping ((M)ATS). Bold *p*-values are significant at α = 0.05 for MANOVA-RM and α_adj_ = 0.007 for ANOVA-RM, respectively. * Multiple outcomes = all seven outcomes combined as a dependent variable. Abbreviations. (M)ATS = (multivariate) ANOVA-type statistic; df1/df2 = degrees of freedom; *p* = *p*-value; TBI = traumatic brain injury; SF-PCS = Short Form (36) Health Survey (physical component score); SF-MCS = Short Form (36) Health Survey (mental component score); PCL-5 = Posttraumatic Stress Disorder Checklist; PHQ-9 = Patient Health Questionnaire; GAD-7 = Generalized Anxiety Disorder questionnaire; QOLIBRI = Quality of Life after Brain Injury; GOSE = Glasgow Outcome Scale—Extended.

**Table 4 jcm-09-01525-t004:** Overview of differences between mTBI groups, time points, and interaction between both effects.

Outcome	mTBI	Time Points	Interaction
SF-36 PCS	--	++	--
SF-36 MCS	++	++	--
PCL-5	--	++	--
PHQ-9	--	++	--
GAD-7	--	--	--
QOLIBRI	++	++	--
GOSE	++	++	--

Note: ++ = significant on α = 0.007, -- not significant. Abbreviations. mTBI = mild traumatic brain injury; SF-PCS = Short Form (36) Health Survey (physical component score); SF-MCS = Short Form (36) Health Survey (mental component score); PCL-5 = Posttraumatic Stress Disorder Checklist; PHQ-9 = Patient Health Questionnaire; GAD-7 = Generalized Anxiety Disorder questionnaire; QOLIBRI = Quality of Life after Brain Injury; GOSE = Glasgow Outcome Scale—Extended.

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
