# Peer review of "Outcomes after Complicated and Uncomplicated Mild Traumatic Brain Injury at Three-and Six-Months Post-Injury: Results from the CENTER-TBI Study"

_jcm, 2020, doi:10.3390/jcm9051525_

Round 1

Reviewer 1 Report

This paper is well-written with validated study method and design although clinical significance would be limited due to simple comparison among groups based on the presence or absence of intracranial abnormal findings. Finding out specific risk factors for the impairment would be much more interesting, however, I think the current form would be acceptable for publication.

Reviewer 2 Report

This is a well-designed study that addresses an important clinical question with direct relevance to clinical practitioners. The manuscript is cleanly organized and well-written. The tables and figures are useful. My only suggestion would be to please more clearly describe how you accounted for the baseline differences in demographic variables (age, working status) between complicated and uncomplicated mTBI groups in your quality of life and functional outcome results.

Also line 240 typo - "after with"

Reviewer 3 Report

My only comment would be to address the implication of post-concussion treatment that does not appear to be either gathered or mentioned in the limitations. 

Also, on line #395, there appears to be an extra word (be). 
